

# Self-concept in poor readers: a systematic review and meta-analysis

Genevieve M. McArthur[1,3], Nicola Filardi[1], Deanna A. Francis[1,3], Mark E. Boyes[2] and Nicholas A. Badcock[1,3,4]

[1] Department of Cognitive Science, Macquarie University, Sydney, New South Wales, Australia
[2] School of Psychology, Curtin University, Perth, Western Australia, Australia
[3] Macquarie University Centre for Reading, Macquarie University, Sydney, New South Wales, Australia
[4] School of Psychological Science, University of Western Australia, Perth, Australia

## ABSTRACT

**Background**. The aims of this systematic review and meta-analyses were to determine if there is a statistically reliable association between poor reading and poor self-concept, and if such an association is moderated by domain of self-concept, type of reading impairment, or contextual factors including age, gender, reading instruction, and school environment.

**Methodology**. We searched 10 key databases for published and unpublished studies, as well as reference lists of included studies, and studies that cited included studies. We calculated standardised mean differences (SMDs) and 95% confidence intervals for one primary outcome (average self-concept) and 10 secondary outcomes (10 domains of self-concept). We assessed the data for risk of bias, heterogeneity, sensitivity, reporting bias, and quality of evidence.

**Results**. Thirteen studies with 3,348 participants met our selection criteria. Meta-analyses revealed statistically significant SMDs for average self-concept ($-0.57$) and five domains of self-concept (reading/writing/spelling: $-1.03$; academic: $-0.67$; math: $-0.64$; behaviour: $-0.32$; physical appearance: $-0.28$). The quality of evidence for the primary outcome was moderate, and for secondary outcomes was low, due to lack of data.

**Conclusions**. These outcomes suggest a probable moderate association between poor reading and average self-concept; a possible strong association between poor reading and reading-writing-spelling self-concept; and possible moderate associations between poor reading and self-concept in the self-concept domains of academia, mathematics, behaviour, and physical appearance.

Corresponding author
Genevieve M. McArthur,
genevieve.mcarthur@mq.edu.au

## INTRODUCTION

The ability to read is a normally-distributed cognitive skill, and hence 16 percent of people have reading skills that fall more than one standard deviation below the level expected for their age or grade (*Shaywitz et al., 1992*). Over the last decade or so, clinicians, teachers, and researchers have become increasingly concerned that people with poor reading are at increased risk for poor emotional health. This concern is supported by
studies reporting statistically significant associations between poor reading and emotional problems (e.g., *Boyes et al., 2016*; *Carroll & Iles, 2006*; *Francis et al., 2019*; *Nelson, Lindstrom & Foels, 2015*).

Despite this evidence, we currently lack a comprehensive theory explaining the mechanisms that link poor reading to emotional problems. As a first step towards developing such a theory, we recently conducted the first systematic review of the associations between poor reading and anxiety and poor reading and depression (*Francis et al., 2019*). We found a statistically significant moderate association between poor reading and anxiety (Cohen's $d = 0.41$), and a weak (yet statistically significant) association between poor reading and depression (Cohen's $d = 0.23$). These results suggest that poor reading is more closely associated with anxiety than depression.

Why might an association exist between poor reading ability and anxiety? Qualitative analyses of interviews with children with poor reading suggest that negative feedback from peers, teachers, and parents may lead them to form a negative self-concept (*Leitao et al., 2017*; *Riddick et al., 1999*). Negative self-concept is known to be a risk factor for anxiety (see *Sowislo & Orth, 2013*, for a review). Thus, poor self-concept may be a candidate mechanism linking poor reading and anxiety.

Self-concept has been defined as a "person's perceptions of him- or herself ... formed through experience with and perceptions of one's environment ... [and] influenced by evaluations by significant others, reinforcements, and attributions for one's own behaviour" (p. 107; *Marsh & Shavelson, 1985*). Current consensus distinguishes between general/global self-concept (which is sometimes called "self-esteem") and specific domains of self-concept. While global self-concept/self-esteem has been defined as "one's global sense of well-being as a person and general satisfaction with oneself" (p. 146; *Zeleke, 2004*), self-concept domains focus on self-perception in specific aspects of life (*Harter, Whitesell & Junkin, 1998*). For example, domains of self-concept include academic self-concept, social self-concept, and athletic self-concept, to name but a few.

Numerous studies have measured general self-concept or domain-specific self-concept in poor readers. The outcomes have been inconsistent, with some studies finding evidence that poor readers have poorer self-concept than typical readers (e.g., *Alexander-Passe, 2006*; *Chapman & Tunmer, 1997*; *Fairhurst & Pumfrey, 1992*) and others failing to find such differences (*Tam & Hawkins, 2012*; *Taylor, Hume & Welsh, 2010*). There are a number of potential explanations for these mixed outcomes. For example, it is possible that poor reading and poor self-concept are not genuinely associated, resulting in findings that are spurious and hence unreliable (Explanation 1). This possibility could be assessed by a systematic review and meta-analyses of well-designed studies that have compared self-concept in groups of people with poor reading to people with typical reading. To our knowledge, no such meta-analysis exists. However, *Chapman (1988)* and *Zeleke (2004)* have done systematic reviews comparing self-concept in groups of people with learning disability who may (or may not) have had poor reading, to control groups with typical development. Chapman found a moderate group effect for general self-concept ($-0.50$; SE $= 0.09$; $N = 21$ studies; Note: a negative effect size indicates poorer scores in an LD group), and a large group effect for academic self-concept ($-0.88$; SE $= 0.16$; $N = 20$ studies).

 

Similarly, Zeleke found that most included studies ($N = 30$) found a reliable difference between groups with and without learning disability for academic self-concept (89%), but not for general self-concept (68%) or social self-concept (70%). Unfortunately, neither of these reviews reported if the overall effect sizes were statistically reliable, or if the groups with learning disability included poor readers. Thus, these reviews do not provide direct evidence for the strength or statistical reliability of the association between poor reading and poor self-concept.

These reviews do, however, suggest a second potential explanation for the mixed outcomes of previous studies. Both reviews found larger group-effect sizes for academic self-concept than general self-concept and social self-concept. This pattern of results suggests that poor reading may be more closely associated with some domains of self-concept (e.g., academic self-concept) than others (e.g., social self-concept). Thus, previous studies of the association between poor reading and poor self-concept may have produced mixed outcomes because poor reading is associated with some domains of self-concept but not others (Explanation 2).

It is also possible that these mixed findings emerged because some studies did not recruit participants with the "right" type of reading problem. People with poor reading have different reading difficulties. Some find it hard to read words accurately via phonological recoding (i.e., the ability to use letter-sound rules to read new words), some with reading words via visual word recognition (or "whole word reading"), and some with reading words via with phonological recoding and visual word recognition (*Castles & Coltheart, 1993*; *McArthur et al., 2013*; *Peterson, Pennington & Olson, 2013*; *Stuart & Stainthorp, 2015*; *Ziegler et al., 2008*). In contrast, some children have no problems with phonological recoding or visual word recognition, but struggle to read texts fluently (*Meisinger, Bloom & Hynd, 2010*) or understand the meaning of texts (*Nation et al., 2010*). It is possible that some of these reading difficulties are more closely associated with poor self-concept than others. Hence, the type of reading problem (or problems) experienced by a sample of poor readers may determine whether or not a study finds an association between poor reading and poor self-concept (Explanation 3).

The strength of such an association, if it exists, may also depend on contextual factors, such as the age of participants, their gender, the type of reading instruction that they have received, and their learning environment. Regarding age, there is evidence that self-concept fluctuates across the lifespan, dropping from childhood to adolescence, increasing throughout adulthood, and then declining in older age (*Marsh, 1989*; *Robins & Trzesniewski, 2005*). There is also evidence that reading self-concept, in particular, starts to decline after the first three years of instruction (*Chapman & Tunmer, 1995*), further supporting the idea that age may modulate the association between self-concept and reading. There may also be effects of gender on self-concept, as suggested by reports of poorer academic self-concept in females than males (*Katzir, Kim & Dotan, 2018*), and increased age-related declines in academic self-concept in girls compared to boys (De Fraine, Van (*De Fraine, Van Dammxe & Onghena, 2007*). Type of reading instruction may also affect the strength of the association between reading self-concept: *Tunmer & Chapman (2002)* and *Chapman & Tunmer (2003)* reported that children who were taught to read

via word-level instruction had higher reading and academic self-concept than children instructed using text-based approaches. More broadly, self-concept—both general and academic—may be modulated by a child's school environment (*Srivastava & Joshi, 2011*; Yaratan & Yucesoylu, 2010). This evidence suggests that contextual factors—including age, gender, reading instruction, and school environment—may determine if a study finds an association between poor reading or not (Explanation 4).

In sum, we currently do not know if poor reading is associated with poor self-concept because of inconsistent findings in the existing literature. These mixed findings might arise for a number of reasons: (1) poor reading is not associated with poor self-concept, producing spurious and unreliable outcomes; (2) poor reading is associated with some types of self-concept but not others; (3) poor self-concept is associated with some types of reading problems but not others; (4) poor reading is association with poor self-concept in some contexts (age, gender, reading instruction, school environment) but not others. The aim of this study was to conduct a systematic review and meta-analysis to determine if there is a reliable association between poor reading and poor self-concept (Explanation 1), and if so, if this association is moderated by domain of poor self-concept (Explanation 2), type of poor reading (Explanation 3), or one or more contextual factors (age, gender, reading instruction, school environment; Explanation 4).

## METHODS

The methods, analyses, and reporting procedures used in this review were guided by the rigorous standards used by Cochrane Reviews to summarise evidence across intervention studies. Minor adjustments were made to the methods to cater for the cross-sectional studies that were included in this review.

### Differences between the registered protocol and review

This review differed from the pre-registered protocol in four respects (*McArthur et al., 2016b*). First, we stated that we would conduct a subgroup analysis to determine if the strength of the association between poor reading and self-concept differs for different types (i.e., subgroups) of self-concept (e.g., reading versus academic versus social versus parent/home). In the current review, a subgroup analysis involved: (1) allocating each accepted study to the appropriate subgroup (e.g., academic self-concept); (2) calculating the mean standardised mean difference (SMD) between poor readers and a control group across all studies in each subgroup; and (3) comparing the SMDs of the subgroups to identify any statistically significant differences. Unfortunately, as explained below, there were not enough studies (i.e., at least 10) to allow us to statistically compare the strength of the associations between poor reading and different domains of self-concept.

Second, we planned to use a second subgroup analysis to determine if poor readers with comorbid impairments (e.g., language or attention problems) are more likely to have poor self-concept than poor readers without comorbid impairments. This was an error in logic since we also aimed to exclude poor readers with comorbid impairments–a common approach used in studies of poor readers used to minimise confounding effects. Thus, this subgroup analysis was not attempted. Third, we planned to search seven sources for grey

literature. We searched these sources to the best of our ability, but our efforts were hampered by poor search tools (opengrey.eu, base-search.net, trove.nla.gov.au, phcris.org.au/roar/, worldcat.org/) and non-relevant content (opendoar.org, research.allacademic.com/). We would not recommend these sources for future systematic reviews.

Fourth, based on the suggestions of a reviewer, we added contextual factors to the review that had not been included in registered protocol (age, gender, reading instruction, and school environment).

## Criteria for considering studies for this review
### Types of studies

This review included studies that compared self-concept in one or more groups of poor readers to appropriate control data. Control data could be provided by a matched group of typical readers or a standardised normative measure. Studies could be cross-sectional studies or intervention studies. In the latter case, data were collected from the initial assessment session prior to intervention. Only studies that used groups of at least 11 participants were included in the review. This (lenient) criterion was calculated from the smallest N needed to detect a very large group effect (Cohens $d = 1.3$) with a power of 0.8 and significance of 0.05 (two-tailed test; AI Therapy Statistics' Sample Size Calculator)

### Types of participants

Participants were English-speaking children, adolescents, or adults whose word reading accuracy or reading fluency was either one grade or year (for children) or one standard deviation (for children, adolescents, and adults) below the mean level of typical readers for no known reason. Specifically, they did not have a comorbid developmental disorder (e.g., autism, language impairment, attention deficit hyperactivity disorder, attention deficit disorder); a physical problem (e.g., impaired vision); or a neurological problem (e.g., brain damage) that could explain their reading difficulty.

This review focused on English-speaking poor readers because English is a non-transparent written language, meaning that many words cannot be read accurately using the letter-sound rules. This contrasts with transparent languages, such as Spanish and Italian, which can be read accurately using the letter-sound rules. The non-transparency of English makes it harder to learn to read than transparent languages, making reading failure more severe and obvious (*Seymour et al., 2003*). Severity of reading failure correlates with academic self-concept (*McArthur et al., 2016a*). Thus, the strength of the relationship between poor reading and self-concept may vary between languages. This review therefore focused on poor readers who spoke English as their primary language at school or work, who lived in a country where English was the official language, and who were receiving reading instruction in English. We did not include studies that included non-English speaking participants who had just arrived in an English-speaking country.

It is noteworthy that the reading criteria used in this review did not include poor reading comprehension on its own (i.e., without evidence of poor reading accuracy or fluency) because poor reading comprehension can arise from poor spoken language comprehension rather than poor reading (*Gough & Tunmer, 1986*). There is evidence that poor spoken language is associated with poor self-concept, raising the risk that poor spoken language,

but not poor reading, could be responsible for an apparent association between poor reading comprehension and poor self-concept.

It is also noteworthy that in line with the latest Diagnostic and Statistical Manual of Mental Disorders (5th Edition), we did not include IQ as a criterion to identify a specific learning problem for reading. We also did not exclude participants based on age, gender, or socioeconomic status (SES) since reading difficulties are experienced by people across all these demographic variables.

### Types of self-concept measures

We only included studies that indexed self-concept with standardised and normed measures that were administered directly to poor readers (i.e., not to carers or teachers). We excluded studies that used indirect self-concept measures administered to significant others since it is difficult for others to estimate a person's true perception of self, and because teachers and peers' perceptions of the academic and social competence of children with learning problems are typically negative (*Kavale & Forness, 1996*). We excluded studies that did not include standardised and normed assessments of self-concept since non-standardised and non-normed measures are less likely to have established reliability and validity than normed assessments, and are less able to reliably indicate if performance falls within or below the average range. If a study included both direct and indirect self-concept measures, or both standardised-normed measures and non-standardised-normed measures, only the direct and standardised-normed indices were included in the analysis.

### Types of outcome measures

*Primary outcomes.* The primary outcome was "average self-concept", which was calculated for each study by taking the mean of scores for all self-concept assessments administered to each group (e.g., poor readers) in that study. This primary outcome was used to test Explanation 1.

*Secondary outcomes.* There are many different domains of self-concept. To identify the most relevant domains for this review, we were guided by the assessments used by the included studies. The second last column of Table 1 shows all these assessments, which could be categorised into 10 domains, as shown in the final column of Table 1: reading/writing/spelling self-concept, academic self-concept, school self-concept, work self-concept, math self-concept, behaviour self-concept, social self-concept, athletic self-concept, physical appearance self-concept, and global self-concept. These secondary outcomes were used to test Explanation 2.

It is noteworthy that the last of these domains - global self-concept - represents the perception of oneself in general, which does not represent a specific domain of self-concept per se. We retained this category as a secondary reliability check for the primary outcome - average self-concept - which indexed the perception of oneself across multiple domains for many studies (see Table 1).

*Timing of outcome measures.* Primary and secondary outcomes were assessed at the same time as reading. We did not include studies that measured reading and self-concept at

McArthur et al. (2020), *PeerJ*, DOI 10.7717/peerj.8772

**Table 1 Summary of methods used in each included study.** Includes categorisation of self-concept measures into secondary outcomes for this review.

| Study | Samples | | Type of self-concept (Explanation 2) | | Type of poor reading (Explanation 3) | Context of poor readers (Explanation 4) |
|---|---|---|---|---|---|---|
| First author **Date** Country | **Poor Readers** **Size** **Ethnicity** **IQ** | **Controls** **Size** **Ethnicity** **IQ** | **Assessment name** Subscale name | **Self-concept category** | **Type** **Criteria** **Assessment** | **Age in Years (range)** **Gender (female/male)** **Reading Instruction** **School Environment** |
| F: Boetsch D: 1996 C: USA (adult study) | S: 18 E: NR IQ: 113 | S: 18 E: NR IQ: 113 | **Adult SP Profile** Adequate Provider General Intellectual Ability Global Self Worth Intimate Relationships Job Competence Math Competence Physical Appearance Reading Competence Spelling Competence Writing Competence | Work Academic Global Social Work Math Physical Appearance Reading Spelling Writing Reading Spelling Writing Reading Spelling Writing | T: NR C: Significant difference between reading/spelling ability and that expected for age and education A: Peabody Individual Achievement Test | A: 45.67 (30-55) G: 0/36 R: NR SE: NR |
| F: Boetsch D: 1996 C: USA (child study) | S: 70 E: NR IQ: 100-107 | S: 67 E: NR IQ: 107-111 | **SP Profile for Learning Disabled Students** Athletic Competence Behavioural Competence Global Self Worth Intellectual Ability Math Competence Physical Appearance Reading Competence Social Acceptance Spelling Competence Writing Competence | Athletic Behavioural Global Academic Math Physical Appearance Reading Spelling Writing Social Reading Spelling Writing Reading Spelling Writing | T: NR C: Significant difference between reading/spelling ability and that expected for age and education and general intelligence A: Gray Oral Reading Test | A: 12.35 (7-18) G: 13/59 R: NR SE: NR |
| F: Chapman D: 2001 C: New Zealand | Reading Recovery: S: 26 E: NR IQ: NR Poor Readers: S: 20 E: NR IQ: NR | S: 80 E: NR IQ: NR | **Reading SC Scale** Reading SC | Reading Spelling Writing | T: NR C: Bottom 20% of school cohort A: Word Identification | Reading Recovery: A: Grade 1-3 G: NR R: Text level SE: NR Poor Readers: A: Grade 1-3 G: NR R: NR SE: NR |

McArthur et al. (2020), *PeerJ*, DOI 10.7717/peerj.8772

**Table 1** (*continued*)

| Study | Samples | | Type of self-concept (Explanation 2) | | Type of poor reading (Explanation 3) | Context of poor readers (Explanation 4) |
|---|---|---|---|---|---|---|
| F: Frederickson<br>D: 2001<br>C: UK | S: 20<br>E: 95% White/2.5% African-Caribbean/ 2.5% Asian<br>IQ: NR | S: 20<br>E: 95% White/2.5% African-Caribbean/ 2.5% Asian<br>IQ: NR | **SP Profile for Children**<br>Athletic Competence<br>Behavioural Competence<br>Global Self Worth<br>Physical Appearance<br>Scholastic Competence<br>Social Acceptance | Athletic<br>Behavioural<br>Global<br>Physical Appearance<br>Academic<br>Social | T: NR<br>C: British Psychological Society definition<br>A: British Abilities Scale 2 (Word Reading) | A: 9.65 (8-11)<br>G: 3/17<br>R: NR<br>SE: Government |
| F: Gold<br>D: 1982<br>C: USA | S: 61<br>E: Mixed<br>IQ: NR | S: 87<br>E: NR<br>IQ: NR<br>(normative data) | **Coopersmith SE Inventory** | Global | T: NR<br>C: Mean grade level far lower (3.54/SD - 2.58) than chronological age<br>A: Wide Range Achievement Test | A: 27.80 (13-71)<br>G: 3/58<br>R: NR<br>SE: Government (prison) |
| F: Holmes<br>D: 2001<br>C: USA | ELS<br>S: 19<br>E: African American<br>IQ: NR<br>Tutoring<br>S: 21<br>E: African American<br>IQ: NR | S: 831<br>E: 88% White/7% Hispanic/1.5% Native American/1.3% Black/Asian 0.7%/Other 1.5%<br>IQ: NR<br>(normative data) | **Perception of Ability Scale for Students** | Global | T: NR<br>C: below 25th percentile on Metropolitan Achievement Test.<br>A: Wide Range Achievement Test 3 | ELS<br>A: 8.5 (7-11)<br>G: NR<br>R: NR<br>SE: Government<br>Tutoring<br>A: 8.5 (7-11)<br>G: NR<br>R: NR<br>SE: Government |
| F: Kerwin<br>D: 1977<br>C: USA | S: 30<br>E: NR<br>IQ: 85+ | S: 44<br>E: NR<br>IQ: 85+ | **Tennessee SC Scale** | Global | T: NR<br>C: Reading age 18+ months below chronological age<br>A: Nelson Reading Test | A: 9th Graders<br>G: NR<br>R: NR<br>SE: Government (military) |

McArthur et al. (2020), *PeerJ*, DOI 10.7717/peerj.8772

**Table 1** (*continued*)

| Study | Samples | | Type of self-concept (Explanation 2) | | Type of poor reading (Explanation 3) | Context of poor readers (Explanation 4) |
|---|---|---|---|---|---|---|
| F: McArthur<br>D: 2016<br>C: Australia | S: 77<br>E: NR<br>IQ: 98 | S: 547<br>E: Stratified<br>IQ: NR<br>(normative data) | **Culture Free SE Inventory 3**<br>Academic SE<br>Home SE<br>Social SE<br>General SE | Academic<br>Home-Parents<br>Social<br>Global | T: NR<br>C: Word reading 1+ SD below chronological age<br>A: Castles and Coltheart 2 (CC2) Nonwords and Irregular Words | A: 1.46 (9–12.5)<br>G: 22/55<br>R: NR<br>SE: NR |
| F: Murray<br>D: 1978<br>C: USA | S: 104<br>E: NR<br>IQ: 95.42 | S: 195<br>E: NR<br>IQ: NR<br>(normative data) | **Piers-Harris Children's SC Scale** | Global | T: NR<br>C: 2+ years delay in reading/writ-ing/spelling despite adequate SES, IQ, schooling<br>A: Wide Range Achievement Test | A: 10.29 (8-15)<br>G: 1:4 ratio<br>R: NR<br>SE: NR |
| F: Palmieri<br>D: 1981<br>C: USA | 15-16<br>S: 16<br>E: 12%+ Black<br>IQ: 85+<br>17-18<br>S: 15<br>E: 12%+ Black<br>IQ: 85+ | 15–16<br>S: 16<br>E: 12%+ Black<br>IQ: 85+<br>17–18<br>S: 16<br>E: 12%+ Black<br>IQ: 85+ | **Rosenberg General SE Scale**<br>McCluskey School SE Scales<br>Academic Ability<br>General Ability SE<br>Peer School SE<br>Teacher School SE | Academic<br>Global<br>Social<br>School | T: NR<br>C: 5th grade or lower on SRA Group Reading Test<br>A: Wide Range Achievement Test | 15-16<br>A: 15–16<br>G: NR<br>R: NR<br>SE: Government<br>17-18<br>A: 17–18<br>G: NR<br>R: NR<br>SE: Government |
| F: Pih<br>D: 1984<br>C: USA | S: 40<br>E: Caucasian<br>IQ: Within 1.5 SD of age mean | S: 40<br>E: Caucasian<br>IQ: Within 1.5 SD of age mean | **Coopersmith SE Inventory** | Global | T: NR<br>C: 1+ Grade below grade level<br>A: Gates-MacGinite Reading Test | A: 7.42-8.58<br>G: 20/20<br>R: NR<br>SE: Government |

McArthur et al. (2020), *PeerJ*, DOI 10.7717/peerj.8772

**Table 1** (*continued*)

| Study | Samples | | Type of self-concept (Explanation 2) | | Type of poor reading (Explanation 3) | Context of poor readers (Explanation 4) |
|---|---|---|---|---|---|---|
| F: Robinson<br>D: 1990<br>C: Australia | S: 44<br>E: NR<br>IQ: 85+ | Normative data<br>S: 831<br>E: 88% White/7% Hispanic/1.5% Native American/1.3% Black/Asian 0.7%/Other 1.5%<br>IQ: 85+ | **Student's Perception of Ability Scale**<br>Academic Ability<br>Arithmetic Ability<br>General Ability<br>Penmanship and Neatness<br>Reading and Spelling<br>School Satisfaction | Academic<br>Math<br>Global<br>Reading Spelling Writing<br>Reading Spelling Writing<br>School | T: NR<br>C: Reading age 3+ years below chronological age<br>A: Neale Analysis of Reading Ability | A: 11.92 (9.08-15.92)<br>G: 11/33<br>R: NR<br>SE: NR |
| F: Taylor<br>D: 2010<br>C: UK | S: 26<br>E: NR<br>IQ: NR | S: 23<br>E: NR<br>IQ: NR | **Culture Free SE Inventory 3**<br>General SE | Global | T: NR<br>C: Reading age more 1+ SD below chronological age<br>A: British Ability Scales 2 | A: 12.5<br>G: 9/17<br>R: NR<br>SE: NR |
different times points (e.g., if self-concept was assessed more than a month/year before or after an individual was assessed for their reading) since this would not valid measure of a current association.

## Search methods for identification of studies
### Electronic searches
We searched the following databases for published studies. The literature search was limited to English-language publications and human participants.
1. MEDLINE (1902 to 2018)
2. PsycINFO (1860 to 2018)
3. EMBASE (1902 to 2018)
4. WILEY
5. PubMed

   We searched the following databases for unpublished studies (grey literature):
1. http://www.opendoar.org
2. http://www.opengrey.eu
3. http://www.base-search.net
4. http://trove.nla.gov.au
5. http://www.phcris.org.au/roar

   We used the following search terms (or the equivalent for unpublished study databases). For poor readers, we used the terms: (1) dyslexia, (2) poor reading, (3) reading disability or difficulty or disorder or impairment or deficit or delay, (4) learning disability or difficulty or disorder or impairment or deficit or delay. For self-concept, we used the terms: (1) self-concept, (2) self esteem, (3) self confidence. For example, the search terms entered into PsycInfo were:
1. (dyslexi* or (poor adj1 read*) or ((read* or learn*) adj1 (dis* or diff* or impair* or def* or delay)) or (word blind*))
2. self and (concept or esteem or confidence)
3. 1 & 2
4. Limit to English Language and Human

### Searching other resources
The reference lists of included studies were reviewed to identify further relevant studies. We also identified and reviewed studies that cited included studies using Google Scholar.

### Data collection and analysis
Data were collected and analysed according to Cochrane Review procedures. All statistics were calculated with Cochrane Review's REVMAN meta-analysis tool.

### Selection of studies
Studies identified by the searches were first checked for duplicates, which were removed. Each study author was paired with another to form a ''review pair'' (GM with DF, NF with NB, MB with NB). Each author in each pair initially screened non-duplicates for eligibility using titles and abstracts. Works that did not include 'reading' or 'dyslexia' were removed

since extensive pilot testing established that such works never include poor readers. Each study author compared their included and excluded studies with their review partner. Any inconsistencies were discussed in detail and until the source of the mismatch was resolved to the satisfaction of both parties. If no agreement could be found, then a referee was used to make a final decision (the first author for review pairs that did not include GM, and NB for review pairs that did include GM). Full-text versions of eligible studies were downloaded and again reviewed by review pairs. Each pair compared accepted and rejected studies, discussed any mismatches, and resolved any inconsistencies. Studies identified via the reference lists and citing studies were also reviewed by two authors, again with any mismatches discussed in person and resolved.

### Data extraction and management

Data were extracted using a customised form included in Appendix S1. The form collected descriptive data (author name, year of publication, reading assessments, any subtests of the assessment, reliability coefficients of assessment, self-concept assessments, any subtests of the assessment, reliability coefficients of assessment) and group data (number of groups in the study, group type, group size, and group means and standard deviations for outcome measures).

Data were extracted by two people. Any inconsistencies between data extracted were discussed and resolved between the pair. Authors of studies were contacted if there was any ambiguity about data (e.g., missing data; see below). A table of correspondence with study authors is included in (Appendix S2). Data was entered into Cochrane's REVMAN by the first author. It was double checked by the second author.

### Dealing with missing data

If a study had missing data (e.g., means, SDs), we requested that data from the corresponding author (see Appendix S2). If this request failed, we contacted the co-authors. If an appeal for missing data did not result in a full data set, we only included data for participants whose results were known.

### Data synthesis

*Multiple groups.* If a study tested multiple groups of poor readers on a particular outcome, we calculated the average mean, SD, and N across these groups before comparing to the mean, SD, and N of the control group. We did the same if a study used multiple groups of controls before comparing to the poor readers.

*Multiple tests.* If a study measured an outcome with more than one assessment that used the same scale (e.g., scaled scores with a mean of 10 and SD of 3), we calculated the average mean and SD across the two assessments. If the assessments used different scales (e.g., one used scaled scores and one used z scores), we (1) used RevMan to calculate the SMDs for each measure separately, (2) calculated the mean SMDs for the two measures, (3) removed the original data entries for the two assessments, and (4) inserted a new entry that used the mean SMD for the experimental group, 0 for the control mean, 1 for the SDs of both groups, and the N of the study.

### Group effects

All studies reported continuous data. Different studies used different assessments to measure outcomes that used different scales (see Table 1 for measures used in each study). We therefore used standardised mean differences (SMDs) with 95% confidence intervals (CIs) calculated from means and SDs for groups with poor reading and typical reading. We used a random effects model to compare SMDs of groups (rather than a fixed effects model) since we predicted that different studies would use different measures to assess self-concept, which would introduce heterogeneity between study outcomes in effect sizes. Random effects models adjust estimates to incorporate heterogeneity more effectively than fixed effects model, which presume similar effects between studies.

We considered SMDs of 0.20, 0.50 and 0.80 to represent small, moderate, and large group effects, respectively (*Cohen, 1992*). In line with *Schünemann et al. (2011)*, we considered 95% CIs to be narrow if the range was around 0.10; medium if the range was around 0.30; and wide if over 0.60. These 95% CI ranges translate to high precision, moderate precision, and low precision in data. We considered group effects with a *P* value less than or equal to 0.05 to be statistically significant and hence statistically reliable.

### Subgroup analyses

Six subgroup analyses were required to determine if there were statistically significant differences between: (1) domains of self-concept (reading/spelling/writing, academic, math, global, behavioural, school, physical appearance, work, social, athletic, and home; Explanation 2); (2) types of reading impairment (phonological dyslexia; surface dyslexia; mixed dyslexia; poor comprehenders; Explanation 3); (3) age groups (children aged up to 12 years; adolescents aged from 13 to 17 years; and adults aged 18 years and above); (4) gender types (female, male); (5) reading instruction types (word level versus text level); and (6) school environments (government school, private school, learning specialist school). In line with Cochrane Review standards, we planned to compare subgroups that comprised at least 10 studies to ensure adequate power (*Deeks, Higgins & Altman, 2011*).

### Risk of bias

We used an adapted version of the Newcastle Ottowa Scale (NOS; (*Wells et al., 2014*) to determine risk of bias in the individual studies (see Table 2 and Appendix S3). We used this scale instead of Cochrane's Risk of Bias procedure because the latter was designed for intervention studies rather than cross-sectional correlational studies, which were the focus of the current review. Two independent authors (GM and NB) made ratings using this scale, which has a maximum of 9 stars/points. Studies were evaluated based on three tiers of ratings: Low (0 to 3 stars); medium (4 to 7 stars); and high (8 to 11 stars). If there was a mismatch between authors, these were discussed and resolved.

### Heterogeneity

We used a $Chi^2$ test with a *P* value of 0.10 to examine the degree of consistency in the effect sizes found by the included studies (i.e., heterogeneity; *Deeks, Higgins & Altman, 2011*). Further, we used the $I^2$ statistic (with a cut-off value of 70%) to estimate the percentage of variance in the effects owing to heterogeneity rather than chance. For any outcome that

McArthur et al. (2020), *PeerJ*, DOI 10.7717/peerj.8772

**Table 2  Risk of bias ratings for each included study.** See Appendix S3 for meaning of a and b ratings, along with allocation of stars (∗). Lower scores represent higher risk of bias (1–4 high risk; 5–7 moderate risk; 8-10 low risk).

| | Sample (Maximum 4 points) | | | | Group comparability | Self-concept assessment | Statistical test | Total | Risk of bias |
|---|---|---|---|---|---|---|---|---|---|
| | Representativeness | Sample size | Response rate | Reading assessment | (Maximum 2 points) | (Maximum 2 points) | (Maximum 2 points) | | |
| Boetsch, Green & Pennington (1996) (adults) | a* | b | a* | a** | a*b* | a* | a* | 8 | Low |
| Boetsch, Green & Pennington (1996) (children) | a* | b | a* | a** | a*b* | a* | a* | 8 | Low |
| Chapman, Tunmer & Prochnow (2001) | a* | b | a* | a** | a* | a* | a* | 7 | Moderate |
| Frederickson & Jacobs (2001) | a* | b | a* | a** | a*b* | a*b* | a* | 8 | Low |
| Gold & Johnson (1982) | a* | b | a* | a** | a* | a*b* | a* | 7 | Moderate |
| Holmes (2001) | a* | b | a* | a** | a* | a*b* | a* | 7 | Moderate |
| Kerwin (1976) | a* | b | a* | a** | a* | a*b* | a* | 7 | Moderate |
| McArthur et al. (2016a) | a* | b | a* | a** | a*b* | a*b* | a* | 8 | Low |
| Murray (1978) | a* | b | a* | a** | a* | a*b* | a* | 7 | Moderate |
| Palmieri (1981) | a* | b | a* | a** | a* | a*b* | a* | 7 | Moderate |
| Pih (1984) | a* | b | a* | a** | a*b* | a* | a* | 8 | Low |
| Robinson & Conway (1990) | a* | b | a* | a** | a* | a*b* | a* | 7 | Moderate |
| Taylor, Hume & Welsh (2010) | a* | b | a* | a** | a*b* | a*b* | a* | 8 | Low |

had an $I^2$ statistic greater than 70%, we (1) double-checked the data, (2) reconsidered the validity and reliability of the measures, and (3) examined outlier studies to see if there was an obvious reason for the outlying result. If something was identified in step (3), we redid the meta-analysis with the offending study removed.

### Sensitivity analysis

We conducted two sensitivity analyses:

1. Removal of any studies with 10 or fewer participants in experimental and control groups
2. Comparison of fixed effects and random effects meta-analyses for outcomes with high heterogeneity.

### Reporting bias

We used funnel plots to explore reporting bias for any outcome that had data from more than 10 studies which did not have similar standard errors for their effect sizes (*Sterne, Egger & Moher, 2011*).

### Quality of evidence

We used a modified version of GRADE (*Schünemann et al., 2011*)-adjusted to suit cross-sectional studies rather than intervention studies - to assess the overall quality of evidence for each outcome. When rating the evidence for each outcome, we started with a high rating. This rating was then downgraded one or two levels (to medium or low) or upgraded one levels for across six factors:

1. Risk of bias: No downgrade (0) if 75% + studies contributing to an outcome are low in majority of biases. Downgrade one level (−1) if 50% to 74% of studies contributing to an outcome are low in majority of biases. Downgrade two levels (−2) if fewer than 50% studies contributing to an outcome are low in majority of biases.
2. Heterogeneity: No downgrade (0) if $I^2$ less than 70% OR $I^2$ greater than 70% but assessment of heterogeneity analysis suggests it did not affect the reliability of results. Downgraded one level (−1) if $I^2 = 70\%$ to 85% and heterogeneity analysis suggests it does affect reliability of results. Downgraded two levels (−2) if $I^2$ greater than 85% and heterogeneity analysis suggests it does affect reliability of results.
3. Indirectness: No downgrade (0) if study directly measures outcomes of interest in the population of interest. Downgraded by one level (−1) if outcome or population are not measured directly. Downgraded two levels (−2) if outcome and population are not measured directly.
4. Imprecision: No downgrade (0) if confidence interval 0 to 0.3. Downgrade on level (−1) if confidence interval 0.3 to 0.6. Downgrade two levels (−2) if confidence interval 0.6 +
5. Publication bias: No downgrade (0) if funnel plot done on more than 10 studies (*Sterne, Egger & Moher, 2011*), and no bias detected. Downgrade one level (−1) if funnel plot cannot be constructed (too few studies) but bias not suspected. Downgrade two levels (−2) if funnel plot not possible (too few studies) and bias suspected.

6.  Other factors: Upgrade one level (+1) if large effect size (0.8+) or no plausible confounds (+1)

### Certainty of outcomes

We interpreted the certainty of each outcome based on *Ryan, Santesso & Hill*'s (*2016*) guide for interpreting the certainty of treatment effects based on GRADE ratings. We modified this guide for use with cross-sectional data, and to take statistical significance into account. For outcomes that were statistically significant and based on high quality of evidence, we concluded that an effect was certain (e.g., a moderate association). For outcomes that were statistically significant and based on moderate quality of evidence, we concluded that an effect was probable (e.g., a probable moderate association). For outcomes that were statistically significant but with low quality of evidence, we concluded that the effect was possible (e.g., possible moderate association). For outcomes that were not statistically significant and had low quality of evidence, we concluded the effect was unlikely (e.g., an unlikely moderate association).

## RESULTS

### Description of studies
#### Results of the search

Figure 1 shows a flow diagram of the search results. Searches of databases of published works identified 6,506 candidate studies. Searches of grey literature identified eight candidate studies. Searches of citations revealed 13 candidate studies. Together the searches revealed 6,527 candidate studies. Removal of duplicate studies resulted in 5,068 candidate studies. Double screening of titles and abstracts of these studies reduced this number to 443. Double examination of the full texts of these studies identified 97 papers. Double review of potential studies from reference lists and citations identified no studies that matched the selection criteria. One study was excluded during the data extraction phase due to lack of accessible data. This left us with 13 accepted studies. Two studies were reported in the same published article (*Boetsch, Green & Pennington, 1996*): one study focused on adults (hereafter *Boetsch, Green & Pennington, 1996* (adults); and one focused on children (hereafter *Boetsch, Green & Pennington, 1996* (children)). Thus the 13 accepted studies were reported in 12 research outputs.

#### Included studies

Thirteen studies with a total of 3,348 participants met the inclusion criteria for this review: *Boetsch, Green & Pennington, 1996* (adults); *Boetsch, Green & Pennington, 1996* (children); *Chapman, Tunmer & Prochnow, 2001*; *Frederickson & Jacobs, 2001*; *Gold & Johnson, 1982*; *Holmes, 2001*; *Kerwin, 1976*; *McArthur et al., 2016a*; *Murray, 1978*; *Palmieri, 1981*; *Pih, 1984*; *Robinson & Conway, 1990*; *Taylor, Hume & Welsh, 2010*. A summary of the methods used by each study is provided in Table 1. For the sakes of brevity in the text and tables, these studies will be referenced using the first author and date (i.e., *Boetsch, Green & Pennington, 1996* (adults); *Boetsch, Green & Pennington, 1996* (children); *Chapman, Tunmer & Prochnow, 2001*; *Frederickson & Jacobs, 2001*; *Gold & Johnson, 1982*;

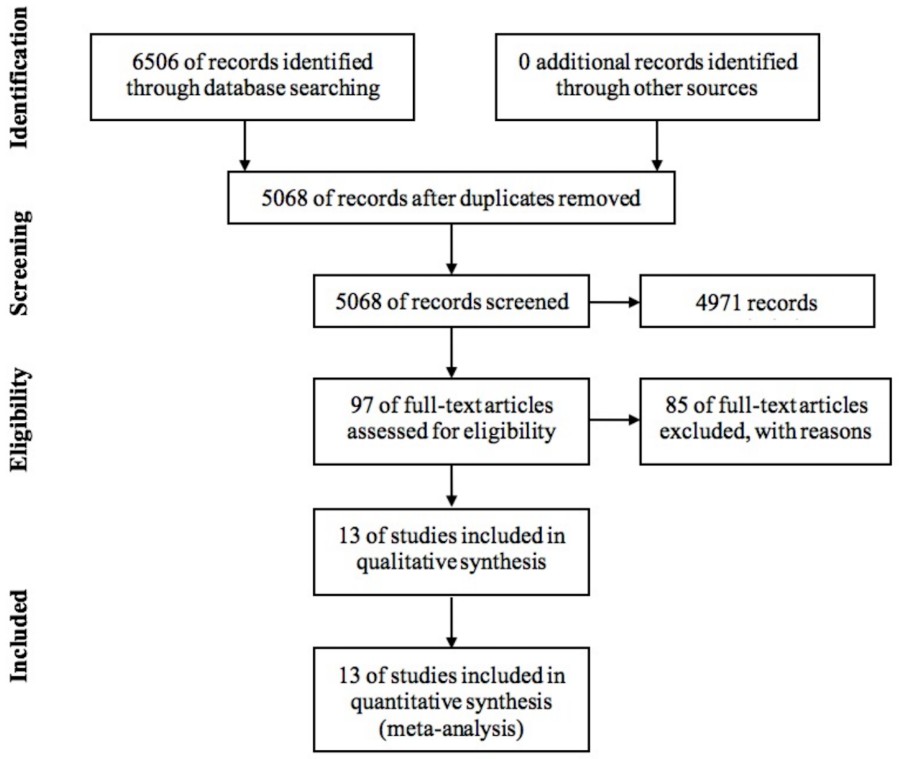

**Figure 1** Flow diagram of the literature search resulting in 13 included studies.

*Holmes, 2001*; *Kerwin, 1976*; *McArthur et al., 2016a*; *Murray, 1978*; *Palmieri, 1981*; *Pih, 1984*; *Robinson & Conway, 1990*; *Taylor, Hume & Welsh, 2010*).

### Study design

All studies compared self-concept scores of poor readers to either a control group (*Boetsch, Green & Pennington, 1996* (adults); *Boetsch, Green & Pennington, 1996* (children); *Chapman, Tunmer & Prochnow, 2001*; *Frederickson & Jacobs, 2001*; *Gold & Johnson, 1982*; *Kerwin, 1976*; *Palmieri, 1981*; *Pih, 1984*; *Taylor, Hume & Welsh, 2010*) or to normed data provided by a standardised test (*Holmes, 2001*; *McArthur et al., 2016a*; *Murray, 1978*; *Robinson & Conway, 1990*).

### Location of studies

Two studies were conducted in Australia (*McArthur et al., 2016a*; *Robinson & Conway, 1990*), one in New Zealand (*Chapman, Tunmer & Prochnow, 2001*), seven in the United States of America (*Boetsch, Green & Pennington, 1996* (adults); *Boetsch, Green & Pennington, 1996* (children); *Gold & Johnson, 1982*; *Holmes, 2001*; *Kerwin, 1976*; *Murray, 1978*; *Palmieri, 1981*; *Pih, 1984*), and two in the United Kingdom (*Frederickson & Jacobs, 2001*; *Taylor, Hume & Welsh, 2010*).

### Participants

Details of the participants in each included study are shown in Table 1.

*Sample size.* Studies fell into two groups: those with similar numbers of participants in the poor-reader and control groups (*Boetsch, Green & Pennington, 1996* (adults); *Boetsch, Green & Pennington, 1996* (children); *Frederickson & Jacobs, 2001*; *Gold & Johnson, 1982*; *Kerwin, 1976*; *Palmieri, 1981*; *Pih, 1984*; *Taylor, Hume & Welsh, 2010*) and those with control groups far larger than the poor-reader group (*Chapman, Tunmer & Prochnow, 2001*; *Holmes, 2001*; *McArthur et al., 2016a*; *Murray, 1978*; *Robinson & Conway, 1990*). The latter pattern stemmed from the use of normative data to represent controls, which is calculated from very large numbers of participants.

*Ethnicity.* Eight studies did not report the ethnicity of their participants (*Boetsch, Green & Pennington, 1996* (adults); *Boetsch, Green & Pennington, 1996* (children); *Chapman, Tunmer & Prochnow, 2001*; *Kerwin, 1976*; *McArthur et al., 2016a*; *Murray, 1978*; *Robinson & Conway, 1990*; *Taylor, Hume & Welsh, 2010*). The samples of three studies comprised a majority of people of caucasian/white ethnicity (*Frederickson & Jacobs, 2001*; *Palmieri, 1981*; *Pih, 1984*). One study focused on readers from African-American backgrounds (*Holmes, 2001*) and one reported that the sample was of mixed ethnicity (*Gold & Johnson, 1982*).

*Intelligence quotient (IQ).* Four studies reported the verbal, non-verbal, or full IQ scores of their participants (*Boetsch, Green & Pennington, 1996* (adults); *Boetsch, Green & Pennington, 1996* (children); *McArthur et al., 2016a*; *Murray, 1978*; *Pih, 1984*). *Kerwin (1976)*, *Palmieri (1981)*, and *Robinson & Conway (1990)* did not report IQ scores but only recruited participants with IQ of 85 or above (i.e., in the average range or above). This information suggests that most poor readers in these studies had IQ scores within the average range at least. The remaining five studies did not use IQ for recruitment and did not report IQ data (*Chapman, Tunmer & Prochnow, 2001*; *Frederickson & Jacobs, 2001*; *Gold & Johnson, 1982*; *Holmes, 2001*; *Taylor, Hume & Welsh, 2010*).

*Reading ability.* The criteria used to recruit poor readers differed between studies. Two studies used a significant difference between actual and expected reading level (*Boetsch, Green & Pennington, 1996* (adults); *Boetsch, Green & Pennington, 1996* (children)); two studies used bottom 20–25% percentile cut-offs for age or grade (*Chapman, Tunmer & Prochnow, 2001*; *Holmes, 2001*); and two studies selected poor readers based on a reading level more than 1 SD below that expected for age (*McArthur et al., 2016a*; *Taylor, Hume & Welsh, 2010*). Some studies used a reading grade that was 1 year (*Pih, 1984*) or 2 grades (*Murray, 1978*), or "far lower" than the expected grade (*Gold & Johnson, 1982*), or was below the 6th grade (*Palmieri, 1981*). Other studies identified poor readers if their reading was more than 18 months (*Kerwin, 1976*) or 3 years (*Robinson & Conway, 1990*) below the age mean. One study used the British Psychological Society criteria, which was word-reading difficulties that were "severe and persistent" (*Frederickson & Jacobs, 2001*). There are three things to note about these criteria: (1) the various criteria well represent the range of criteria used to identify poor readers in reading research; (2) the criteria used by a study did not determine its inclusion in this review, which had its own criteria for reading (see Participants above); and hence (3) data presented by all these studies showed that the

reading scores of these samples fell more than one SD below the level expected for their age.

*Age.* 11 of the 13 included studies tested children aged between 7 and 18 years (*Boetsch, Green & Pennington, 1996* (children); *Frederickson & Jacobs, 2001*; *Holmes, 2001*; *McArthur et al., 2016a*; *Murray, 1978*; *Palmieri, 1981*; *Pih, 1984*; *Robinson & Conway, 1990*; *Taylor, Hume & Welsh, 2010*) or grades 1 to 9 (*Chapman, Tunmer & Prochnow, 2001*; *Kerwin, 1976*). Two studies tested a broader age group: 13 to 71 years (*Gold & Johnson, 1982*) and 30 to 55 years (*Boetsch, Green & Pennington, 1996* (adults)).

*Gender.* The majority of studies tested a higher proportion of males than females (*Boetsch, Green & Pennington, 1996* (children); *Frederickson & Jacobs, 2001*; *Gold & Johnson, 1982*; *McArthur et al., 2016a*; *Murray, 1978*; *Robinson & Conway, 1990*; *Taylor, Hume & Welsh, 2010*). Indeed, one study only assessed males (*Boetsch, Green & Pennington, 1996* (adults)). One study included an equal number of females and males (*Pih, 1984*). The remaining four studies did not report numbers of females and males (*Chapman, Tunmer & Prochnow, 2001*; *Holmes, 2001*; *Kerwin, 1976*; *Palmieri, 1981*).

*Reading instruction.* Only one study reported the type of reading instruction received by poor readers (*Chapman, Tunmer & Prochnow, 2001*). The remaining 12 studies did not (*Boetsch, Green & Pennington, 1996* (adults); *Boetsch, Green & Pennington, 1996* (children); *Frederickson & Jacobs, 2001*; *Gold & Johnson, 1982*; *Holmes, 2001*; *Kerwin, 1976*; *McArthur et al., 2016a*; *Murray, 1978*; *Palmieri, 1981*; *Pih, 1984*; *Robinson & Conway, 1990*; *Taylor, Hume & Welsh, 2010*).

*School environment.* Most studies recruited participants from government schools (Frederickson, 2010; *Holmes, 2001*; *Kerwin, 1976*; *Palmieri, 1981*; *Pih, 1984*; *Taylor, Hume & Welsh, 2010*) or a government prison (*Gold & Johnson, 1982*). No school reported recruitment from privately-funded schools. Six studies did not report the types of schools from which participants were recruited (*Boetsch, Green & Pennington, 1996* (adults); *Boetsch, Green & Pennington, 1996* (children); *Chapman, Tunmer & Prochnow, 2001*; *McArthur et al., 2016a*; *Murray, 1978*; *Robinson & Conway, 1990*).

### Domains of self-concept
Four of the studies measured self-concept for reading/spelling/writing (*Boetsch, Green & Pennington, 1996* (adults); *Boetsch, Green & Pennington, 1996* (children); *Chapman, Tunmer & Prochnow, 2001*; *Robinson & Conway, 1990*). Six studies measured academic self-concept (*Boetsch, Green & Pennington, 1996* (adults); *Boetsch, Green & Pennington, 1996* (children); (*Frederickson & Jacobs, 2001*; *McArthur et al., 2016a*; *Palmieri, 1981*; *Robinson & Conway, 1990*). Three studies assessed math self-concept (*Boetsch, Green & Pennington, 1996* (adults); *Boetsch, Green & Pennington, 1996* (children); *Robinson & Conway, 1990*), twelve measured global self-concept (*Boetsch, Green & Pennington, 1996* (adults); *Boetsch, Green & Pennington, 1996* (children); (*Frederickson & Jacobs, 2001*; *Gold & Johnson, 1982*; *Holmes, 2001*; *Kerwin, 1976*; *McArthur et al., 2016a*; *Murray, 1978*;

*Palmieri, 1981*; *Pih, 1984*; *Robinson & Conway, 1990*; *Taylor, Hume & Welsh, 2010*), two behavioural self-concept, (*Boetsch, Green & Pennington, 1996* (children); *Frederickson & Jacobs, 2001*) two school self-concept (*Palmieri, 1981*; *Robinson & Conway, 1990*), three physical appearance self-concept (*Boetsch, Green & Pennington, 1996* (adults); *Boetsch, Green & Pennington, 1996* (children); *Frederickson & Jacobs, 2001*), one work self-concept (*Boetsch, Green & Pennington, 1996* (adults)), five social self-concept (*Boetsch, Green & Pennington, 1996* (adults); *Boetsch, Green & Pennington, 1996* (children); *Frederickson & Jacobs, 2001*; *McArthur et al., 2016a*; *Palmieri, 1981*), two athletic self-concept (*Boetsch, Green & Pennington, 1996* (children); (*Frederickson & Jacobs, 2001*), and one home self-concept (*McArthur et al., 2016a*).

### Types of reading impairment

As is shown in Table 1 (Poor-Reader Type column), no study reported poor reader's type of reading difficulty. Examination of the tests used to assess the reading skills of participants suggests that all samples had a combination of different reading difficulties.

### Outcome measures

The measures used by each study to measure primary and secondary outcomes are shown in Table 1 (see Self-Concept Assessment column). Measures used to assess the primary outcome (average self-concept) and secondary outcomes (different domains of self-concept) include the Adult Self Perception Profile (*Boetsch, Green & Pennington, 1996* adults), the Self Perception Profile for Learning Disabled Children (*Boetsch, Green & Pennington, 1996* children),  the Reading Self Concept Scale (*Chapman, Tunmer & Prochnow, 2001*), *Harter*'s (*1985*) Self-perception Profile for Children (*Frederickson & Jacobs, 2001*), the Coopersmith Self Esteem Inventory (*Gold & Johnson, 1982*; *Pih, 1984*), *Boersma & Chapman*'s (*1992*) Perception of Ability Scale for Students (*Holmes, 2001*), the Tennessee Self Concept Scale (*Kerwin, 1976*), the Culture-Fair Self Esteem Inventory (*McArthur et al., 2016a*; *Taylor, Hume & Welsh, 2010*), the McCluskey School Self Esteem Scales and Rosenberg General Self Esteem (*Palmieri, 1981*), *Chapman & Boersma*'s (*1979*) Students Perception of Ability Scale (*Robinson & Conway, 1990*), and the Piers-Harris Children's Self-Concept Scale (*Murray, 1978*).

### Funding

Of the 12 studies included in this review, four declared funding support from independent funding organisations: the Australian Research Council (Australia; *McArthur et al., 2016a*), the Department of Health Education and Welfare (US; *Gold & Johnson, 1982*), the Massey University Research Fund (New Zealand; *Chapman, Tunmer & Prochnow, 2001*), the New Zealand Ministry of Education (*Chapman, Tunmer & Prochnow, 2001*), the National Health and Medical Research Council (Australia; *McArthur et al., 2016a*), and the NICHD, NIMH, and NIH (US, *Boetsch, Green & Pennington, 1996*).

### Excluded studies

Appendix S4 lists studies that reading researchers might expect to be included in this review, but were excluded because they did to meet our review criteria. Reasons include:

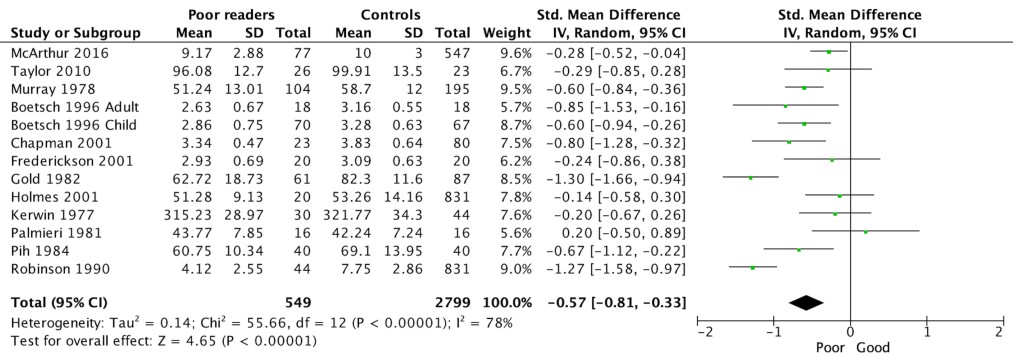

**Figure 2** Forest plot of data for the primary outcome (average self-concept).

the study did not assess participants on a quantitative reading test; the study did not include a self-concept assessment with known validity or reliability; the study only assessed reading using a measure of reading comprehension; the study recruited poor readers with comorbid problems the study focused on poor readers that did not speak English.

### Primary outcome

*Group effect.* The outcomes of the random effects model for the primary outcome are shown in Fig. 2 and Table 3. The SMD for average self-concept was calculated from 13 studies and 3,348 participants. The number of domains from which average self-concept was calculated varied between studies: 10 domains for two studies (*Boetsch, Green & Pennington, 1996*-adults; *Boetsch, Green & Pennington, 1996*-children); six domains for two studies (*Frederickson & Jacobs, 2001*; *Robinson & Conway, 1990*); five domains for one study (*Palmieri, 1981*); four domains for one study (*McArthur et al., 2016a*); and one domain for five studies (*Chapman, Tunmer & Prochnow, 2001*; *Holmes, 2001*; *Kerwin, 1976*; *Pih, 1984*; *Murray, 1978*; *Taylor, Hume & Welsh, 2010*). The SMD for average self-concept was −0.57 (95% CI [−0.81 to −0.33]; $Z = 4.65$; $P < 0.001$). Note that a negative effect size for self-concept indicates poorer scores in poor readers.

*Risk of bias in included studies.* Table 2 shows the results of the risk of bias assessments for each study included for the primary outcome. All studies were rated as truly or somewhat representative of the average in the target population. The sample size of the majority of studies was not justified. All bar one study used a standardized reading assessment with data reported; only one study did not. All studies used English poor readers, and around half controlled for additional factors such as attention, age, sex, SES, neurological or medical problem. All studies used a normed index of self-concept, many read items aloud to the participants, and many provided self-report data in addition to parent or teacher report. The total risk of bias scores indicated that five studies had high scores (low risk), and seven studies had medium scores (medium risk). No studies had low scores (high risk).

McArthur et al. (2020), *PeerJ*, DOI 10.7717/peerj.8772

**Table 3  Summary of findings (random effects model).** Includes size of effects, heterogeneity, quality of evidence (GRADE) and conclusion about the certainty of each outcome.

| Outcome (self-concept) | Studies | Sample size | Random effects model | | | Heterogeneity | | | | GRADE | Certainty of outcome |
|---|---|---|---|---|---|---|---|---|---|---|---|
| | | | SMD (95% CI) | Z | P | Chi² | df | P | I² | | |
| **Primary Outcome** | | | | | | | | | | | |
| Average | 13 | 3,348 | −0.57 [−0.81 to −0.33] | 4.65 | <.001 | 55.66 | 12 | <.001 | 78 | Medium | Probable moderate association |
| **Secondary Outcomes** | | | | | | | | | | | |
| Reading-Spelling-Writing | 5 | 2,002 | −1.03 [−1.66, −0.41] | 3.23 | <.001 | 41.69 | 4 | <.001 | 90 | Low | Possible large association |
| Academic | 7 | 2,595 | −0.67 [−0.97, −0.36] | 4.26 | <.001 | 21.84 | 6 | .001 | 73 | Low | Possible moderate association |
| Math | 4 | 1,899 | −0.64 [−1.03, −0.24] | 3.17 | .002 | 10.89 | 3 | .01 | 72 | Low | Possible moderate association |
| Global | 9 | 2,046 | −0.57 [−0.87, −0.28] | 3.79 | <.001 | 37.21 | 8 | <.001 | 78 | Low | Possible moderate association |
| Behavioural | 2 | 177 | −0.32 [−0.62, −0.03] | 2.15 | .03 | 0.07 | 1 | .78 | 0 | Low | Possible moderate association |
| School | 2 | 910 | −0.28 [−1.29, 0.74] | 0.05 | .59 | 7.26 | 1 | .007 | 86 | Low | Unlikely moderate association |
| Physical Appearance | 3 | 213 | −0.28 [−0.55, −0.01] | 2.03 | .04 | 1.43 | 2 | .49 | 0 | Low | Possible moderate association |
| Work | 1 | 36 | −0.23 [−0.89, 0.42] | 0.70 | .48 | | | | | Low | Unlikely moderate association |
| Social | 5 | 869 | −0.15 [−0.34, 0.04] | 1.57 | .12 | 4.33 | 4 | .36 | 8 | Low | Unlikely small association |
| Athletic | 2 | 177 | 0.15 [−0.15, 0.44] | 0.98 | .33 | 0.92 | 1 | .34 | 0 | Low | Unlikely small association |
| Home | 1 | 1,804 | 0.31 [−0.29, 0.91] | 1.01 | .31 | | | | | Low | Unlikely moderate association |

**Table 4  Outcomes of the fixed effects analyses for the primary and secondary outcomes.**

| Outcome (self-concept) | Studies | Sample size | Fixed effects model | | | Heterogeneity | | | |
|---|---|---|---|---|---|---|---|---|---|
| | | | SMD (95% CI) | Z | P | Chi² | df | P | I² |
| **Primary** | | | | | | | | | |
| Average | 13 | 3,348 | −0.61 [−0.72, −0.50] | 11.35 | <.001 | 55.66 | 12 | <.001 | 78 |
| **Secondary** | | | | | | | | | |
| Reading-Spelling-Writing | 5 | 2,005 | −1.13 [−1.32, −0.95] | 12.08 | <.001 | 42.60 | 4 | <.001 | 91 |
| Academic | 7 | 2,598 | −0.69 [−0.83, −0.55] | 9.39 | <.001 | 22.30 | 6 | .001 | 73 |
| Math | 4 | 1,902 | −0.70 [−0.90, −0.51] | 7.19 | <.001 | 11.23 | 3 | .01 | 73 |
| Global | 9 | 2,049 | −0.68 [−0.81, −0.55] | 10.42 | <.001 | 37.83 | 8 | <.001 | 79 |
| Behavioural | 2 | 177 | −0.32 [−0.62, −0.03] | 2.15 | .03 | 0.07 | 1 | .78 | 0 |
| School | 2 | 910 | −0.59 [−0.86, −0.32] | 4.23 | <.001 | 7.26 | 1 | .007 | 86 |
| Physical Appearance | 3 | 213 | −0.28 [−0.55, −0.01] | 2.03 | .04 | 1.43 | 2 | .49 | 0 |
| Work | 1 | 36 | −0.23 [−0.89, 0.42] | 0.70 | 0.48 | | | | |
| Social | 5 | 869 | −0.15 [−0.33, 0.02] | 1.73 | .08 | 4.33 | 4 | .36 | 8 |
| Athletic | 2 | 177 | 0.15 [−0.15, 0.44] | 0.98 | 0.33 | 0.92 | 1 | 0.34 | 0 |
| Home | 1 | 1,804 | 0.31 [−0.29, 0.91] | 1.01 | 0.31 | | | | |

*Heterogeneity.* The heterogeneity for the primary outcome was greater than 70% and statistically significant (Chi² = 55.66; *df* = 12; P <.001; I² = 78%). Thus, we (1) double-checked the data, (2) reconsidered the validity and reliability of the measures, and (3) examined outlier studies to see if there was an obvious reason for the outlying result. The last step revealed a single study with a positive effect for average self-concept (0.20; (*Palmieri, 1981*). When we removed this study from the SMD calculation it strengthened the SMD somewhat (−0.62; 95% CI [0.-0.86 to −0.38]; Z = 5.02; P < 0.001)) but the heterogeneity was not reduced (Chi² = 50.35; *df* = 11; P < 0.001; I² = 78%).

*Sensitivity.* Since all studies had more than 10 participants, the sensitivity analysis involved comparing our planned random effects analysis to a fixed effects analysis (see Table 4). The SMD for average self-concept was stronger than the random effects model (−0.61; 95% CI [−0.72 to −0.50]; Z = 11.35; P <.001) but the heterogeneity remained exactly the same (Chi² = 55.66; *df* = 12; P < 0.001; I² = 78%). This sensitivity analysis and the heterogeneity analysis suggested that the primary outcome effect size was reliable despite the heterogeneity, hence we based our conclusions on the random-effects model, since it adjusts estimates to incorporate heterogeneity (*Deeks, Higgins & Altman, 2011*).

*Reporting bias.* Since our primary outcome had data for more than 10 studies ( *N* = 13 studies) which had varying standard errors, we examined the primary outcome for reporting biases using a funnel plot (see Fig. 3). While this plot did not show the approved inverted funnel shape (i.e., studies with greater precision cluster more closely around the SMD than studies with less precision), neither did it illustrate asymmetry due to (1) an absence of imprecise studies with small SMDs, or (2) a preponderance of imprecise studies with large

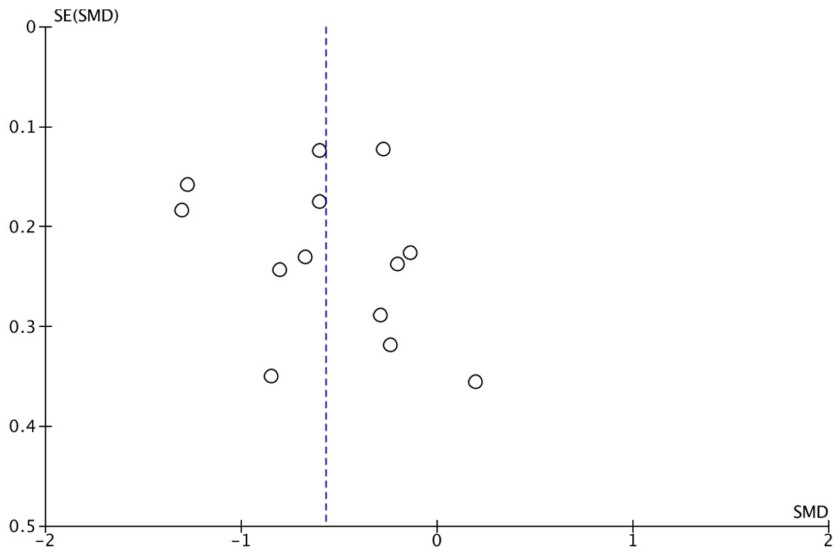

**Figure 3** **Funnel plot of the relationship between standard mean differences (SMDs) and standard errors (SEs) for studies contributing to the primary outcome (average self-concept).** Vertical dotted line represents mean SMD for all studies.

SMDs. Thus, data available - albeit limited - does not suggest reporting bias for the primary outcome.

*Quality of evidence.* The GRADE ratings for average self-concept is shown in Table 5. The quality of evidence for the primary outcome was reduced to moderate by imprecision due to a large confidence interval [−0.81 −0.33].

*Certainty of outcome.* Based on the quality of evidence ratings, SMDs, and statistical significance, we concluded that there is a probable moderate association between poor reading and average self-concept.

### Secondary outcomes

*Group effect.* The number of studies contributing to the secondary outcomes varied considerably, ranging from global self-concept (10 studies) to work and home self-concept (a single study each; see Table 1). The same was true for number of participants, ranging from 2595 participants (academic self-concept; Note: participants include normative sample) to 36 participants (work self-concept).

Figure 4 shows the SMDs (with confidence intervals) for each secondary outcome. In decreasing order of strength, the SMDs the self-concept domains were: reading-spelling-writing self-concept (−1.03; 95% CI [−1.66 to −0. 41]); academic self-concept (−0.67; 95% CI [−0.97 to −0.36]); math self-concept (−0.64; 95% CI [−1.03 to −0.24]); global self-concept (−0.57; 95% CI [−0.87 to −0.28]); behavioural self-concept (−0.32; 95% CI [−0.62 to −0. 03]); school self-concept (−0.28; 95% CI [−1.29 to 0.74]); physical appearance self-concept (−0.28; 95% CI [−0.55 to −0. 01]); work self-concept (−0.23; 95% CI [−0.89 to 0.42]); social self-concept (−0.15; 95% CI [−0.34 to 0.04]); athletic

**Table 5   Quality of evidence (GRADE) rating table.** For each outcome, the initial rating is high. This was increased or decreased according to the ratings of six factors (see following notes). The final rating is high, medium, or low quality of evidence, which defines the certainty of each outcome, which is based on the guidelines of *Ryan, Santesso & Hill (2016)*. The following criteria were used to calculate the ratings (*McArthur et al., 2018*, Table 6): "Note. 1. Risk of bias: No downgrade (0) if 75% + studies contributing to an outcome are low in majority of biases. Downgrade one level (−1) if 50% to 74% of studies contributing to an outcome are low in majority of biases. Downgrade two levels (−2) if fewer than 50% studies contributing to an outcome are low in majority of biases. 2. Heterogeneity: No downgrade (0) if $I^2$ less than 70% or $I^2$ greater than 70% but assessment of heterogeneity and sensitivity analyses suggest the outcome is reliable. Downgraded one level (−1) if $I^2$ 70% to 85% and heterogeneity and sensitivity analyses suggest that it does affect reliability of results. Downgraded two levels (−2) if $I^2$ greater than 85% and heterogeneity and sensitivity analyses suggest it does affect reliability of results. 3. Indirectness: No downgrade if study directly measures outcomes of interest in the population of interest. Downgraded by one level if outcome or population are not measured directly. Downgraded two levels (−2) if outcome and population are not measured directly. 4. Imprecision: No downgrade (0) if confidence interval 0 to 0.3. Downgrade one level (−1) if confidence interval 0.3 to 0.6. Downgrade two levels (−2) if confidence interval 0.6 +. 5. Publication bias: No downgrade (0) if funnel plot done on more than 10 studies (*Sterne, Egger & Moher, 2011*), and no bias detected. Downgrade one level (−1) if funnel plot cannot be constructed (too few studies) but bias not suspected. Downgrade two levels (−2) if funnel plot not possible (too few studies) and bias suspected. 6. Other factors: Upgrade one level (+1) if large effect size (0.8+) or no plausible confounds.

| Self-concept Outcome | Risk of Bias[1] | Heterogeneity[2] | Indirectness[3] | Imprecision[4] | Publication bias[5] | Other[6] | GRADE | Certainty of outcome |
|---|---|---|---|---|---|---|---|---|
| **Primary outcome** | | | | | | | | |
| Average | 0 | 0 | 0 | −1 | 0 | 0 | Medium | Probable |
| **Secondary outcomes** | | | | | | | | |
| Reading-Spelling-Writing | 0 | 0 | 0 | −2 | −1 | +1 | Low | Possible |
| Academic | 0 | 0 | 0 | −2 | −1 | 0 | Low | Possible |
| Math | 0 | 0 | 0 | −2 | −1 | 0 | Low | Possible |
| Global | 0 | 0 | 0 | −1 | −1 | 0 | Low | Possible |
| Behavioural | 0 | 0 | 0 | −1 | −1 | 0 | Low | Possible |
| School | 0 | 0 | 0 | −1 | −1 | 0 | Low | Possible |
| Physical Appearance | 0 | 0 | 0 | −1 | −1 | 0 | Low | Possible |
| Work | 0 | 0 | 0 | −2 | −1 | 0 | Low | Possible |
| Social | 0 | 0 | 0 | −1 | −1 | 0 | Low | Possible |
| Athletic | 0 | 0 | 0 | −1 | −1 | 0 | Low | Possible |
| Home | 0 | 0 | 0 | −2 | −1 | 0 | Low | Possible |

self-concept (0.15; 95% CI [−0.15 to 0.44]); and home self-concept (0.31; 95% CI [−0.29 to 0.91]). This order was interesting since it suggested that poor reading is most closely related to self-concept domains that are related to academia (e.g., reading/spelling/writing, academic, math).

*Risk of bias.* Studies contributing to each secondary outcome were the same as those to the primary outcome. Hence, the studies contributing to each secondary outcome were a mix of low and medium risk of bias (see Table 2).

*Heterogeneity.* The heterogeneity of each secondary outcome is shown in Table 3. Heterogeneity was higher than 70% for five of these outcomes (reading-spelling-writing, academic, math, global, and school), so we again (1) double-checked the data, (2) reconsidered the validity and reliability of the measures, and (3) examined outlier studies to see if there was an obvious reason for the outlying result for each outcome. The last step suggested the removal of *Holmes (2001)* from reading/writing/spelling self-concept (SMD = -1.30; 95% CI [−1.75 to −0.85]); $Z = 5.66$; $P < .00001$; $X^2 = 12.72$; $df = 3$; $P = .005$; $I^2$

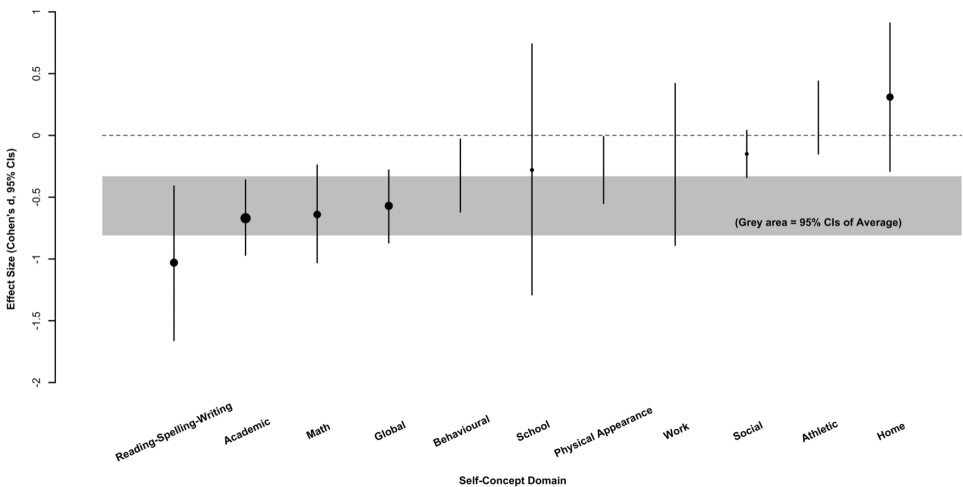

**Figure 4** SMDs (with mean confidence interval; CI) for each secondary outcome in decreasing order of strength (left to right).

= 76%); *Holmes (2001)* and *Palmieri (1981)* from academic self-concept (SMD = −0.85; 95% CI [−1.14 to −0. 56; $Z = 5.75$; $p < .00001$; $X^2 = 11.06$; $df = 4$; $P = .03$; $I^2 = 64\%$); and *Palmieri (1981)* from global self-concept (SMD = −0.63; 95% CI [−0.93 to −0. 33]; $Z = 4.10$; $P < .0001$; $X^2 = 33.44$; $df = 7$; $P < .0001$; $I^2 = 79\%$). The SMDs increased somewhat but the heterogeneity remained high.

*Sensitivity.* Since all studies had more than 10 participants, the sensitivity analysis involved comparing our planned random effects analysis to a fixed effects analysis (see Table 4). The SMDs for the latter were the same or somewhat higher than the random effects analysis, and the heterogeneity remained the same. This suggested that the effect sizes were reliable despite the heterogeneity.

*Reporting bias.* No secondary outcome had data from more than 10 studies and so none were examined for reporting bias.

*Quality of evidence.* The GRADE ratings for the different domains of self-concept are shown in Table 5. The quality of evidence for all these outcomes was low, primary due to imprecision of data (i.e., large confidence intervals) and because there were not enough studies to assess any of these outcomes for publication bias.

*Certainty of outcomes.* Based on the quality of evidence ratings, SMDs, and statistical significance, we concluded that there was a possible strong association between poor reading and reading-writing-spelling self-concept. We also concluded that there was a possible moderate association between poor reading and self-concept in the academic, mathematic, global, behavioural, and physical appearance domains. Due to low quality of evidence, the small and moderate associations between poor reading and school, work, social, athletic, and home self-concept were concluded to be unlikely.

### Subgroup analyses

*As outlined above, six subgroup analyses were required for this review.* In line with Cochrane guidelines, we planned to compare subgroups if they comprised at least 10 studies. None of the subgroups included this minimum number of studies. It is noteworthy that the heterogeneity of the outcomes for the subgroups with the largest number of studies (9 and 7 for global and academic self- concept, respectively) was high (i.e., $I^2$ greater than 70%). This review therefore lacked the power and reliability required for any subgroup analyses.

## DISCUSSION

### Summary of main results

Inconsistent findings in the existing literature obscure whether poor reading is associated with poor self-concept. In this systematic review, which identified 13 studies comprising 3,348 participants, we examined four explanations for these mixed findings: poor reading is not associated with poor self-concept (Explanation 1); poor reading is only associated with certain types of self-concept (Explanation 2); poor self-concept is only associated with certain types of reading problems (Explanation 3); and poor reading is more strongly associated with poor self-concept in some contexts more than others (Explanation 4).

A meta-analysis of the primary outcome data revealed that the association between poor reading and average self-concept was statistically significant and moderately strong. The reliability of this finding was supported by the association between poor reading and global self-concept, which was almost identical in size. These findings suggest a probable moderate association between poor reading and average self-concept, which fails to support Explanation 1 as an explanation for the mixed findings in the literature.

Subsequent meta-analyses of the secondary outcomes revealed that the association between poor reading and reading/writing/spelling self-concept was statistically significant and large, and that there were statistically significant and moderate associations between poor reading and self-concept domains of academia, mathematics, behaviour, and physical-appearance. In contrast, the evidence for associations between poor reading and self-concept domains of school, work, social life, athletics, and home was of poor quality. A lack of studies (i.e., at least ten per subgroup) prevented a statistical comparison of the associations between poor reading and different self-concept domains, which were planned to address Explanation 2. However, ranking these associations in order of strength suggested that poor reading was most closely associated with the domains of self-concept that focus most on reading and academia (i.e., reading-spelling-writing, academia, math). More studies of sufficient quality are needed to test this suggestion statistically.

Unfortunately, no study reported the specific type or types of poor reading that challenged the poor readers in their samples. Similarly, all bar one study failed to report on the type of reading instruction received by poor readers. The majority of included studies did report on the age, gender, and school environment of their poor-reading samples, but no subgroup within these contextual factors comprised 10 studies, prohibiting statistical comparisons between subgroups. Thus, we could not assess if poor self-concept is associated with some types of reading impairment and not others (Explanation 3). Nor could we

assess if the contextual factors of age, gender, reading instruction, or school environment influenced the strength of the association between poor reading and poor self-concept. Many more quality studies are required to determine if type of reading impairment or contextual factors affect the association between poor reading and poor self-concept.

### Overall completeness and applicability of evidence

The outcomes of the 13 studies in this review appear applicable to English-speaking poor readers for at least four reasons. First, studies were conducted in each of the major English-speaking countries in the world - specifically (in alphabetical order), Australia (two studies), New Zealand (one study), the USA (seven studies), and the UK (two studies).

Second, eight of the nine studies that reported statistics for gender recruited more males than females. This is representative of studies of poor readers in general (*Miles, Haslum & Wheeler, 1998*). Some researchers claim that this recruitment bias reflects a higher incidence of reading difficulties in boys than girls (e.g., *Miles, Haslum & Wheeler, 1998*). However, others have suggested that more boys than girls are recruited for studies because (1) boys with poor reading are more likely to misbehave when they are frustrated or bored than girls, and hence their failure is more apparent (*Shaywitz et al., 1990*); and (2) societies are more concerned about the academic success of boys than girls, raising awareness of failure in boys relative to girls (*Sadker & Sadker, 2010*). These suggestions are supported by studies reporting that girls and boys are equally likely to have poor reading (e.g., *Shaywitz et al., 1990*), and that girls and boys do not differ in their reading-related cognitive processes (e.g., *Jiménez et al., 2011*). Thus, while this review is representative of a gender recruitment bias in studies of poor readers, this bias needs to be avoided in future studies.

Third, many, but not all, poor readers in the included studies were reported to have IQ scores within or above the mean range. This reflects the type of poor reader who gains the most attention in reading research, namely, people with poor reading despite average intelligence (a condition that has been referred to as "specific reading disability" or "developmental dyslexia"). However, as mentioned in the Methods, IQ is no longer used as a diagnostic criterion for learning difficulties in reading. Thus, the outcomes of this review are applicable to poor readers with various levels of IQ.

Finally, four studies in this review recruited adult poor readers, and nine studies recruited children with poor reading. This is representative of research on poor reading, which typically focuses on children. However, many children with poor reading carry their reading challenges into adulthood. Hence, it would be helpful if more studies of adult poor readers included measures of self-concept and its domains.

## Quality of the evidence

As shown in Table 5, the quality of evidence in this review was based on five factors. The first was risk of bias. As illustrated by Table 2, all studies had a low-risk judgement for the majority of the biases assessed in this review. The second factor was heterogeneity (see Table 3), which was high (i.e., above 70%) for the primary outcome (average self-concept) and five of the 11 secondary outcomes. This was unsurprising given the limited number of studies that met the basic research criteria required for studies of poor readers. To

determine the degree to which this heterogeneity may compromise the reliability of each outcome, we conducted heterogeneity and sensitivity analyses. The outcomes suggested that the outcomes with high heterogeneity were indeed reliable, and hence this did not compromise the quality of evidence of our outcomes.

The third factor was the directness of measures, which could not compromise the results because this review's criteria dictated that only studies using direct assessments of both reading and self-concept were included. In contrast, the fourth factor (imprecision) did affect the quality of the results because all outcomes had confidence intervals that were rated as wide or very wide. The fifth factor—reporting bias—was not an issue for the primary outcome (average self-concept). However, it was an unknown factor for the secondary outcomes which did not have enough studies to produce valid funnel plots.

In sum, the quality of evidence for this review was supported by analyses of risk of bias (low to moderate), heterogeneity (existent but not a threat to reliability), and reporting bias (for the primary outcome), but was challenged somewhat by imprecision (i.e., large confidence intervals) and by unknown reporting bias for secondary outcomes.

## Potential biases in the review process

There are three reasons why there appeared to be minimal bias affecting the results of this review. First, the funnel plot of the outcome with the requisite number of studies ($N > 10$; average self-concept) suggested no evidence of reporting bias or bias owing to outliers. Second, a comparison of effects using fixed- and random-effects analyses revealed very similar results for all primary and secondary outcomes, suggesting statistical reliability. Third, sensitivity analyses for all outcomes produced similar results to the original analyses, again supporting the reliability of the outcomes.

## Agreements and disagreements with other studies or reviews

One aim of this review was to determine if there was conflicting evidence for poor self-concept in poor readers because there is no reliable association between poor reading and poor self-concept (Explanation 1). The outcomes of this review failed to support this hypothesis, instead finding a statistically-significant moderate association between poor reading and average self-concept and global self-concept. These outcomes favour previous studies that found an association between poor reading and poor self-concept (e.g., *Alexander-Passe, 2006*) over those who did not (e.g., *Tam & Hawkins, 2012*).

At the same time, this review validates the conflict between studies that did and did not find poor self-concept in poor readers, since the secondary outcomes (i.e., the different domains of self-concept) produced inconsistent findings. Ranking these domains in order of strength of association with poor reading revealed that this association was strongest for domains of self-concept that focused on self-perceptions relating to reading and academia (i.e., reading/writing/spelling, academic, math). While this result may appear utterly predictable, it was not predicted by numerous studies in this review that did not assess poor readers for self-concept domains related to reading or academia (see Table 1). This finding provides preliminary support for the idea that the existing literature comprises mixed findings about self-concept in poor readers because poor reading is more closely associated with some types of self-concept than others (Explanation 2).

## Implications for research and theory

In terms of future research, this review revealed that many more studies are needed to understand the association between poor reading and self-concept. It is important that these studies actually test the reading skills of their poor readers to confirm that they have reading problems. We were surprised by the number of studies that we had to exclude because participants were not actually tested objectively and/or recently for their reading ability.

It is noteworthy that we also excluded a handful of studies that assessed self-concept with non-standardised measures with unknown reliability and validity. It might be argued that this decision was too stringent, given that some non-standardised and non-validated assessments comprise items similar to standardised and validated measures. Our decision was guided by the same principle as the exclusion of studies that did not test participants' reading: maximising quality of data. Now that a statistically reliable association between poor reading and poor self-concept has been supported by good quality data, it might be of use to see if the same results emerged including studies that used non-validated self-concept measures employing similar items to validated measures.

Another observation made during this review was the number of studies that used a general measure of self-concept without measuring specific domains of self-concept. Given the review outcomes, which suggest that poor reading is most strongly related to self-concept in the domains of reading and academia, we would suggest that future research consider different domains of self-concept in addition to, or even rather than, a global or average measure of self-concept.

We were not surprised to find that the studies included in this review failed to report the type of reading impairments that characterised their sample of poor readers. Unfortunately, a small minority of reading studies recruit or report on the specific types of poor reading that characterise the poor readers in their samples. Future research focusing on any aspect of poor readers - including their self-concept—would do well to report this information.

It would also be helpful if future research conducted reviews like this for other languages. For reasons outlined under Types of Participants above, this review focused solely on English-speaking poor readers. It would be interesting to see if similar reviews done in different languages produced different outcomes, since these may provide clues about the mechanisms responsible for an association between poor reading and poor self-concept. For example, if this association was significantly weaker in poor readers who read a language that was quicker and easier to learn than English (e.g., Italian), we might hypothesise that poor reading is more obvious in English than Italian, which may lead to more negative feedback about poor reading in English than Spanish, and hence poorer self-concept.

Finally, as outlined in the Introduction, one theoretical impetus of this review was to determine if self-concept might be a mechanism linking poor reading to anxiety, which we found to be moderately and statistically-significantly associated with poor reading in a previous systematic review (*Francis et al., 2019*). The current review similarly found a statistically-significant moderate association between poor reading and poor self-concept. Whilst these moderate and significant associations are by no means evidence for a causal relationship between poor reading, poor self-concept, and anxiety, these associations do

support the further investigation of self-concept as a potential factor linking poor reading to anxiety. We are currently conducting a case series intervention study and a cross-sectional study to explore this possibility.

## CONCLUSIONS

In sum, this review assessed four possible explanations for the mixed evidence for an association between poor reading and poor self-concept: (1) poor reading is not reliably associated with poor self-concept; (2) poor reading is associated with some types of self-concept but not others; (3) poor self-concept is associated with some types of reading impairment but not others; and (4) the strength of the association between poor reading and poor self-concept may be affected by contextual factors such age, gender, reading instruction, and school environment. The outcomes of this review and meta-analyses failed to support the first explanation: there was a statistically-significant moderate association between poor reading and average self-concept as well as global self-concept. The outcomes provided preliminary support for the second explanation: self-concept in domains of reading and academia were more strongly associated with poor reading than other domains. Unfortunately, due to lack of reporting or lack of studies, this review was unable to assess if type of reading impairment or contextual factors (age, gender, reading instruction, school environment) influence the strength of the association between poor reading and poor self-concept.

### Funding

One author (Deanna A. Francis) on this manuscript received a Macquarie University Research Excellence Scholarship (MQRES). The funders had no role in study design, data collection and analysis, decision to publish, or preparation of the manuscript.

### Grant Disclosures

The following grant information was disclosed by the authors:
Macquarie University Research Excellence Scholarship (MQRES).

### Competing Interests

Genevieve McArthur, Nicholas Badcock, and Mark E. Boyes are Academic Editors for PeerJ.

### Author Contributions

- Genevieve M. McArthur conceived and designed the experiments, performed the experiments, analyzed the data, prepared figures and/or tables, authored or reviewed drafts of the paper, and approved the final draft.
- Nicola Filardi performed the experiments, prepared figures and/or tables, authored or reviewed drafts of the paper, and approved the final draft.

- Deanna A. Francis performed the experiments, prepared figures and/or tables, authored or reviewed drafts of the paper, and approved the final draft.
- Mark E. Boyes performed the experiments, authored or reviewed drafts of the paper, and approved the final draft.
- Nicholas A. Badcock conceived and designed the experiments, performed the experiments, prepared figures and/or tables, authored or reviewed drafts of the paper, and approved the final draft.

### Data Availability

This is a systematic review and meta-analysis which does not include raw data.

### Supplemental Information

Supplemental information for this article can be found online at http://dx.doi.org/10.7717/peerj.8772#supplemental-information.

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
