# Peer review of "Self-concept in poor readers: a systematic review and meta-analysis"

_PeerJ, doi:10.7717/peerj.8772_

## Round 0.1 · original submission · Major Revisions

I have now received two reviews from experts in the field. As you will read, both reviewers are very positively inclined toward your work, and I share their overall impression. I thank the reviewers for their work.

The reviews are clear and constructive; please refer to them for details. I think that for a successful revision you will need to clarify and justify better the choice of the selected papers, why you decided to confine yourself to a limited number of papers and also the possible effects on the results due to multiple factors that vary across studies (see comments of both reviewers).

I am looking forward to your revised version.

·

Basic reporting

The report is well written. It refers to literature appropriately and uses a professional article structure. It has relevant results to the hypotheses.

Experimental design

The research question is well defined, relevant and meaningful. The investigation is rigorous and well conducted. The methods are described with sufficient detail.

Validity of the findings

As this is a meta-analysis, underlying data are provided in other publications. Analysis is on means and standard deviations from each sample, which are provided within the paper.

Additional comments

The authors state that there is inconsistent research about the relationship between poor reading and self concept and provide three possible explanations for this: the null hypothesis that there is no relationship; that the relationship depends upon what type of reading difficulty is measured; or that the relationship depends on which type of self-concept is measured. While these are three possible explanations, it seems to me that the authors ignore the highly likely explanation that the relationship between the two variables depends on multiple contextual factors which vary across studies, including the age of participants, the environment in which the participants are placed, and the type of reading tuition received. The authors are not in a position to assess these contextual factors, for the most part. However, I feel that the role of additional contextual factors should be acknowledged in the introduction and discussion.

Reviewer 2 ·

Basic reporting

.

Experimental design

.

Validity of the findings

.

Additional comments

I liked this paper a lot. It was beautifully written, clear and concise. There are a lot of meta-analysis details that requires knowledge on this type of an analysis but the authors give references for meta-analysis guidelines etc.

In my mind there is only one bigger issue; the small amount of studies included in the analysis. Because of that some interesting subgroup analyses were not possible. On one hand this weakens the paper but on the other hand it also brings forth the need for more studies on this topic. I'm not sure if the choice to omit papers with non-validated or -standardized papers is the best one in this case. I can understand why the authors did this but I wonder if some of the omitted papers would have been of high quality despite this flaw. Self-concept is, in the end, assessed with few questions only and some papers may have used almost equal items to the validated scales? It would be interesting to see if the results would change after entering the studies with non-standardized measures in the analysis. However, I do understand the importance of reliability and validity of measures used and the authors choice in this matter.

More minor issues;
-Line 98: "proffer" should be offer
-Beginning of methods; here it would be helpful to be clearer about what is "subgroup" as you also compare poor readers to others. It is little difficult to follow what exactly are the subgroup analyses here.
-Beginning of method, line 134, sentence starting "First,": This was a difficult sentence to follow.
-line 301: "knew" misplaced?
-Line 311-312: what intervention effects? I guess hat is not the topic of this paper..?
- Authors call self-concept questionnaire items "a test". It seems a little pompous?
- Please note in the limitations that only English-speaking samples were analyzed.

---

## Round 0.2 · accepted · Accept

I am happy to inform you that your article has been accepted for publication on PeerJ.

Reviewer 2 ·

Basic reporting

no comment

Experimental design

no comment

Validity of the findings

no comment

Additional comments

Wonderful work. Thanks for writing this paper. I hope this will inspire many new studies in this important area!

(please note that on the second para in Methods it says "see XXX", which likely needs to be re-phrased...)